# Abnormal Programmed Cell Death of Tapetum Leads to the Pollen Abortion of *Lycium barbarum* Linnaeus

**Xin Zhang [1,2,3], Zhanlin Bei [2,3], Haijun Ma [2,3], Zhaojun Wei [2], Jun Zhou [2,3], Yufeng Ren [2,3], Wendi Xu [2,3], Peng Nan [1], Yuguo Wang [1], Linfeng Li [1], Wenju Zhang [1], Ji Yang [1], Yang Zhong [1] and Zhiping Song [1,*]**

[1] Ministry of Education Key Laboratory for Biodiversity Science and Ecological Engineering, Institute of Biodiversity Science, School of Life Sciences, Fudan University, Shanghai 200438, China
[2] College of Biological Science and Engineering, North Minzu University, Yinchuan 750021, China
[3] Innovation Team for Genetic Improvement of Economic Forests, North Minzu University, Yinchuan 750021, China
[*] Correspondence: songzp@fudan.edu.cn

**Abstract:** Programmed cell death (PCD) in tapetum provides nutrients for pollen development. Once this process becomes abnormal, the pollen will be aborted, and sterile lines will be formed. *Lycium barbarum* L. is a well-known medicinal and edible plant, and male sterile lines play an important role in the cultivation of its new varieties by utilizing hybrid vigor. However, the male sterility mechanism of *L. barbarum* is poorly understood. In this study, the main characteristic changes during the pollen development of *L. barbarum* sterile line (Ningqi No. 5) and fertile line (Ningqi No. 1) were compared through anatomical observation, physiological detection and gene expression analysis. Anatomical observation showed that compared with that of the fertile line, the tapetum of the sterile line persisted during pollen development, the organelle morphology and number of tapetal cells changed remarkably, and the PCD was remarkably delayed. Membranous peroxidation caused by reactive oxygen species (ROS) in the sterile line occurred from the late tetrad to the pollen grain stage, and that in Ningqi No. 1 transpired in the tetrad stage. This difference in the dynamic changes of ROS affected the redox balance of microspore development. qPCR detection of *DYT1* and *MS1* genes regulating tapetum development showed that compared with those in the fertile line, the expression levels of both genes in the sterile line changed significantly from pollen mother cell stage to pollen grain stage. This finding may be associated with the start-up delay of tapetal PCD. All these results suggested that abnormal tapetal PCD is an important mechanism leading to male sterility in *L. barbarum*.

**Keywords:** *Lycium barbarum*; reactive oxygen species; male sterility; tapetum; programmed cell death

## 1. Introduction

The analysis of pollen development is of great significance to obtain an in-depth understanding of the mechanism of male sterility in promoting hybrid breeding and utilizing hybrid vigor. *Lycium barbarum* is an important medicinal and edible species native to northern China. This plant is difficult to crossbreed because of its infinite inflorescence and small flowers, which hinder artificial emasculation. The male sterile line can greatly improve hybrid breeding, which is beneficial to the cultivation of new varieties and the utilization of hybrid vigor [1]. However, the sterile male line of *L. barbarum* is rare, and its sterility mechanism is poorly understood. Ningqi No. 5 is the only sterile line of *L. barbarum* and is a natural mutant of the self-compatible fertile line Ningqi No. 1. Ningqi No. 5 has excellent characteristics, such as large fruit size, good taste, and stable and high yield, and it has broad application prospects for hybridization. However, no conclusive studies have been conducted to reveal its inheritance and cytogenetic basis.

The tapetum is a material and energy source for developing microspores. It is adjacent to the microspores and provides them with nutrients in the pollen sac [2,3]. Sporopollenin

and the exine proteins of pollen grains are produced through programmed cell death (PCD) [4]. Therefore, the development of tapetum and the normality of PCD are directly related to male fertility. The timely formation and degradation of the tapetum are necessary for the formation of fertile pollen and are regulated by the spatiotemporal expression of tapetal PCD genes [2,3,5], including DYSFUNCTIONAL TAPETUM1 *(DYT1)* [6], DEFECTIVE IN TAPETAL DEVELOPMENT AND FUNCTION1 *(TDF1/MYB35)* [7], ABORTED MICROSPORES *(AMS)* [8], MS188/MYB80 [9], and MALE STERILITY1 *(MS1)* [10,11]. The tapetum regulatory pathway *DYT1-TDF1-AMS-MS188-MS1* is composed of these genes and is strictly controlled by an evolutionarily conserved transcriptional cascade [12,13]. The mutation or abnormal expression of any gene in this regulatory pathway can lead to pollen abortion and male sterility [14]. ROS, an important signaling molecule in cellular PCD, mediates tapetal PCD and leads to tapetum degradation [15,16]. Low ROS level leads to the enlargement and delayed degeneration of tapetal cells in rice, resulting in pollen abortion and male sterility. This process is associated with the mutation of *DTC1*, which regulates tapetal PCD by inhibiting ROS scavenging activity [17]. The ROS level in normal rice anthers increases at developmental stages 8 and 9 and decreases at stage 11. By contrast, this ROS level change disappears in abortive anthers (*mads3* mutant), and tapetal PCD occurs earlier, suggesting that ROS level changes are associated with tapetal PCD [18]. Whether abnormal tapetum development is the cause of pollen abortion for the sterile line of *L. barbarum* and is related to ROS level is unknown.

To understand the mechanism of *L. barbarum*, we investigated the tapetum development and microspore abortion of Ningqi No. 5 from histological, physiological, and molecular biology perspectives. Paraffin sections and transmission electron microscopy (TEM) were used to examine the histomorphological and dynamic changes of the tapetum and its cell organelles during the development of Ningqi No. 5 microspores. Apoptosis during microspore development was analyzed by terminal deoxynucleotidyl transferase-mediated 2-deoxyuridine 5-triphosphate nick labeling (TUNEL). Changes in physiological indices during the development of microspores and changes in the vital gene expression levels of tapetum regulatory pathways were also measured. The results showed that the dynamic level of ROS affects the initiation and progression of tapetal PCD and correlates with the expression of genes in the conserved tapetum regulatory pathway *DYT1-TDF1-AMS-MS188-MS1*, resulting in abnormal tapetum development and microspore abortion. These findings advance our understanding of infertility mechanisms.

## 2. Materials and Methods

### 2.1. Plant Materials

A 7-year-old *L. barbarum* inbred line, Ningqi No. 1 (fertile), and its natural mutant, Ningqi No. 5 (sterile), were planted in Yuxin *L barbarum* Ciyuan, Ningxia, China (106°4′56″ E, 38°29′46″ N), with a row spacing of 1.5 m and a plant spacing of 0.5 m and under routine management. The test site has an altitude of 1084 m, a mid-temperate continental climate, an average annual temperature of 8.5 °C, an average annual precipitation of 203 mm, and an average annual relative humidity of 60%.

### 2.2. Sampling Methods

During the germination period of flower buds from 2019 to 2020, 15 individuals of sterile and fertile lines, respectively, were randomly collected for the cytological observation of anther development. The development of microspores was divided into five stages according to the length of the flower buds: archesporial cell stage (Ar, ≤1.3 mm), sporogenous cell stage (Sp, 1.31–1.89 mm), pollen mother cell stage (Pm, 1.90–2.93 mm), tetrad stage (Te, 2.94–4.0 mm), and pollen grain stage (Po, 4.01–5.0 mm). The anthers at different developmental stages were quickly stripped, and sections were prepared for observation.

## 2.3. Paraffin Section Preparation for Histomorphological Observation

In order to investigate the histomorphology of microspore abortion, we collected the flower buds of Ningqi No. 5 and Ningqi No. 1 at five development stages. The paraffin section was carried out with reference to Zhou et al. (2018) [19], including the following steps. (1) Fixation: flower buds were fixed in 50% formalin acid–ethanol solution for 24 h at a solid-to-liquid ratio of 25:1. According to the corresponding relationship between the anther development period and the length of the flower bud discovered by our earlier research, 30 flower buds were measured and fixed at each of the five development periods. (2) Dehydration: the concentration of dehydrating agent was divided into 6 grades (75% alcohol for 4 h, 85% alcohol for 2 h, 90% alcohol for 2 h, 95% alcohol for 1 h, absolute alcohol I for 30 min, absolute alcohol II for 30 min). (3) Transparent: xylene was used twice in this step for 10 min each time. In the process of transparency, if a white mist appeared around the material, the water in the material had not been removed completely, and it needed to be returned to pure alcohol for dehydration, followed by transparency again. (4) Wax immersion: supersaturated wax was added to the material containing xylene. The wax at 65 °C was replaced three times, each time soaking for one hour. (5) Embedding: the melted wax was placed into the embedding frame; the tissue was removed from the dehydration box and then placed into the embedding frame before the wax solidified. (6) Section: the embedded wax was cut into a neat trapezoid with a slice thickness of 4 μm with (LEICA RM205). (7) Dewaxing: The slices were placed into xylene I for 20 min, xylene II for 20 min, absolute ethanol I for 5 min, absolute ethanol II for 5 min, alcohol for 5 min, and washed with tap water. (8) Staining: The sections were dyed in safranine staining solution for 2 h, then decolorized with alcohol, stained in Fast Green staining solution for 6–20 s, and dehydrated with absolute alcohol. (9) Seal: the slice was placed into xylene transparent for 5 min, and neutral gum was used as a sealant. (10) Microscopic examination, image acquisition, and analysis were observed by a digital scanner (Pannoramic MIDI, 3DHISTECH, Budapest, Hungary) [20].

## 2.4. Ultramicroscopic Observation of Tapetal Cells

A total of 30 flower buds of Ningqi No. 5 and Ningqi No. 1 were collected at the Pm and Te stages and fixed in 3% glutaraldehyde solution for more than 24 h at a solid-to-liquid ratio of 25:1. After being re-fixed with 1% osmium tetroxide solution, the samples were dehydrated by acetone and embedded in Ep812 [21]. Semithin sections were stained with toluidine blue; ultrathin sections (70 nm) were cut using a diamond knife (DiATOME, Nidau, Switzerland), stained with uranyl acetate and lead citrate, and observed using a TEM instrument (JEM-1400FLASH, JEOL, Tokyo, Japan) [22]. We observed the wall layer of the pollen sac, the morphology of the pollen mother cells and organelles, and anther development status.

## 2.5. TUNEL Assay

The TUNEL assay was performed using the paraffin block of the paraffin sections prepared during histomorphological observation. The paraffin block was deparaffinized twice in xylene solution, repaired in DNase-free proteinase K solution, and washed with PBS. Afterward, the samples were placed in the reaction solution, incubated with primary and secondary antibodies, counterstained with DAPI, and mounted for observation. All reagents were purchased from Servicebio, Wuhan (China). The sections were observed under a fluorescence microscope (Eclipse C1, Nikon, Tokyo, Japan) and an imaging system (DS-U3, Nikon, Tokyo, Japan) [23].

## 2.6. Determination of $H_2O_2$, $O_2{}^-$, and Malondialdehyde (MDA) Contents and Antioxidative Enzyme Activities

The contents of $H_2O_2$, $O_2{}^-$, and MDA and the activities of antioxidant enzymes GST, POD, APX, and SOD were measured in the anthers collected at each of the five developmental stages of Ningqi No. 5 and Ningqi No. 1, by using commercially available kits

(Soleibao Biological Co., Ltd., Shanghai, China) in accordance with the manufacturer's instructions. $H_2O_2$ and titanium sulfate form yellow titanium peroxide complex, which has characteristic absorption at 415 nm [24,25]. The detection of $O_2^-$ is described by the hydroxylamine hydrochloride oxidation method [26,27]. MDA adopts the thiobarbituric acid method and its maximum absorption peak is at 532 nm. All the measurements were conducted in triplicate and repeated every year.

### 2.7. Determination of the Expression Levels of Genes Regulating Tapetum Development

The total RNA of the different developmental stages of stamen in the two strains was extracted using Trizol reagent (Tiangen, Beijing, China), and RNA quantity and quality were analyzed by measuring the absorbance at 260 nm using a spectrophotometer (ALLSHENG, Hangzhou, China). Reverse transcription polymerase chain reaction (RT-PCR) and quantitative real-time PCR (qRT-PCR) were used to study the expression of the genes involved in tapetum development-related pathways. NCBI was searched for the homologous genes of *DYT1* and *MS1*. Degenerate primers were designed on the basis of these genes with reference to the primer design of *DYT1* and *MS1* by Gu [28] and Lu [29] (Supplementary Material Table S2). These designed primers were used to perform qPCR on the RNAs of Ningqi No. 5 and Ningqi No. 1 at different developmental stages. For qRT-PCR, cDNA was first synthesized by RT-PCR with HiScript Q Select RT SuperMix on a Biorad IQ5 PCR system (Biorad IQ, Hercules, CA, USA). The qRT-PCR program was as follows: pre-deformation at 95 °C for 3 min and 40 cycles of 95 °C for 10 s, 60 °C for 30 s, and 72 °C for 15 s. Actin gene was used as an internal control. The expression level of each gene was normalized using the $2^{-\Delta\Delta Ct}$ method. Each experiment was repeated three times.

### 2.8. Statistical Analysis

Physiological indices (ROS and enzymatic analyses) and gene expression were compared using a paired T-test between Ningqi No. 5 and Ningqi No. 1, and significance and correlation analyses were assessed using SPSS software (IBM, Almond, NY, USA). Data were presented as the mean $\pm$ standard deviation of three biological replicates.

## 3. Results

### 3.1. Histomorphological Characteristics of Ningqi No. 1 and Ningqi No. 5

We found that compared with those of Ningqi No. 1, the filaments of Ningqi No. 5 are longer and placed above the stigma, and the anthers are shrunken with no pollen attached to the surface. No significant difference in tapetum morphology at the Ar and Sp stages was observed between Ningqi No. 5 and Ningqi No. 1. The anther wall layer of both lines is composed of four layers, with the outermost layer being the epidermis, followed by endothecium (fiber layer, layers 2–3), middle layer (layer 1), and tapetum layer (layer 1) (Figure 1A,F). From the Pm stage, with the expansion of the pollen sac cavity, the tapetum of Ningqi No. 1 moves close to the pollen mother cell and shrinks inward (Figure 1G). At this time, the tapetal cells are ellipsoid with one or two dark nuclei and a dense cytoplasm. The pollen mother cell, which is located in the center of the pollen sac, is large, and the nucleus and cytoplasm are darkly stained. In Ningqi No. 5, the tapetal cells are neither close to the pollen mother cells nor shrink inward; instead, they are close to the middle layer (Figure 1B). These cells are oblong, lightly stained, and have precise edges. During the Te stage, the intercellular junctions of the tapetum of Ningqi No. 1 are loose, and the edges are unclear. The cells have formed granule bodies and began to exhibit apoptotic characteristics. The inner tetrad edge is fuzzy, and only a small amount of callose can be found outside (Figure 1H). At this time, the tapetum of Ningqi No. 5 is still close to the middle layer and has dark staining and clear cell edges; no granular bodies were observed, indicating the lack of apoptosis signs. The inner tetrad is surrounded by thick callose (Figure 1C). At the Po stage, the anther wall of Ningqi No. 1 only has the epidermis, an inner wall (thickened fiber layer), and a middle layer; the tapetum layer is completely decomposed (Figure 1I,J). The pollen grains are round or oval and plump, and

the nuclear material is stained. Meanwhile, the anther wall of Ningqi No. 5 is composed of epidermis (vacuole), an inner wall, a middle layer, and a complete tapetum. Its pollen grains are irregularly shriveled with some missing contents (Figure 1D,E) (Table S1). All these findings indicated that abnormal tapetal PCD may be related to the pollen abortion of Ningqi No. 5.

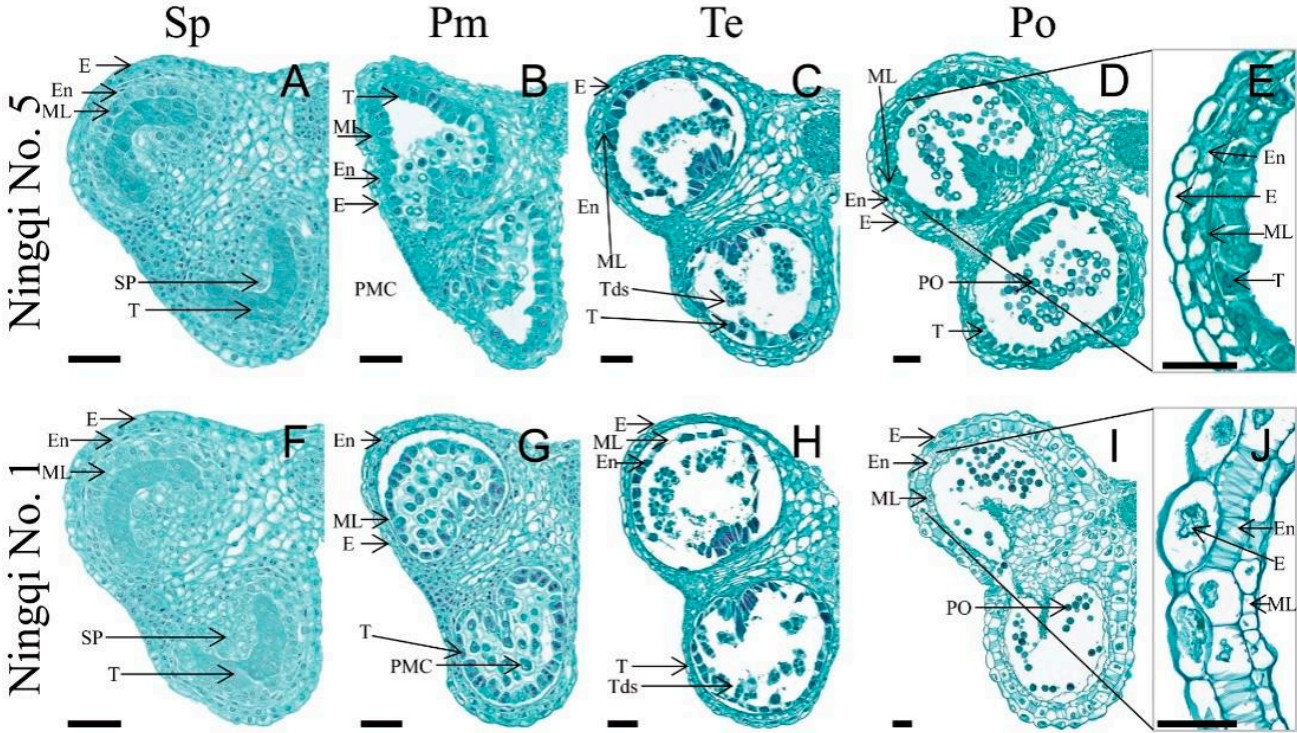

**Figure 1.** Paraffin sections of Ningqi No. 5 and Ningqi No. 1 pollen sacs at different development stages. (**A–E**) Pollen sac development of Ningqi No. 5; (**F–J**) pollen sac development of Ningqi No. 1; (**A,F**) pollen sacs of Sp stage; (**B,G**) pollen sacs at Pm stage; (**C,H**) pollen sacs at Te stage; (**D,I**) pollen sacs at Po stage; (**E,J**) anther wall layers at Po stage. E, epidermis; EN, endothecium; ML, middle layer; T, tapetum; Tds, tetrad; PMC, pollen mother cell; SP, sporogenous cell; Po, pollen grain. Scale bar = 50 μm.

At the cellular level, we found that pollen abortion significantly differed between the sterile and fertile lines and was initiated at the Pm and Te stages. To reveal the changes at the subcellular level in these two crucial stages, we compared the ultrastructure of the tapetum and pollen mother cells from the two developmental stages of Ningqi No. 5 and Ningqi No. 1. At the Pm stage, four layers of parietal cells were differentiated from the anthers of the two strains (Figures 2A and 3A). Compared with Ningqi No. 1, Ningqi No. 5 has a narrower and longer epidermal layer, a large vacuole in tapetal cells that occupies most of the intracellular space, and a nucleus that is small and squeezed to one side of the cell (Figure 2B). Meanwhile, the tapetum of Ningqi No. 1 has a large nucleus, dense mass, and a few small vacuoles (Figure 3B). The pollen mother cells of Ningqi No. 5 are quadrangular, slightly flat, and contain few organelles (e.g., mitochondria and endoplasmic reticulum) and a prominent vacuole (Figure 2D,E). In addition to normal mitochondria, heteromorphic mitochondria were found in the pollen mother cells of Ningqi No. 5— "Fuzzy onion, Fzo" (Figure S1). Compared with those of Ningqi No. 5, the microsporocytes of Ningqi No. 1 are more polygonal and slender; they contain more organelles, but no vacuoles and more lipid droplets (Figure 3D,E). At the Te period, the anther wall of Ningqi No. 1 has three layers, the tapetum shows signs of apoptosis, apoptotic bodies are formed, and the rest of the wall layers are all vacuolated (Figure 3F,G). The pollen mother cells at this point have thick cell walls (Figure 3H) and are rich in organelles and lipid droplets

(Figure 3I). Meanwhile, the anther wall layer of Ningqi No. 5 still has four intact layers. The tapetal cells shrink but maintain good integrity; they contain a large number of lipid droplets and a dense cytoplasm (Figure 2G,H). The tetraspores are enclosed by callose, and the intracellular vacuoles are severely vacuolated (Figure 2I). These findings are consistent with histological observations. Therefore, the preservation of tapetum in Ningqi No. 5 may be the reason for its pollen abortion.

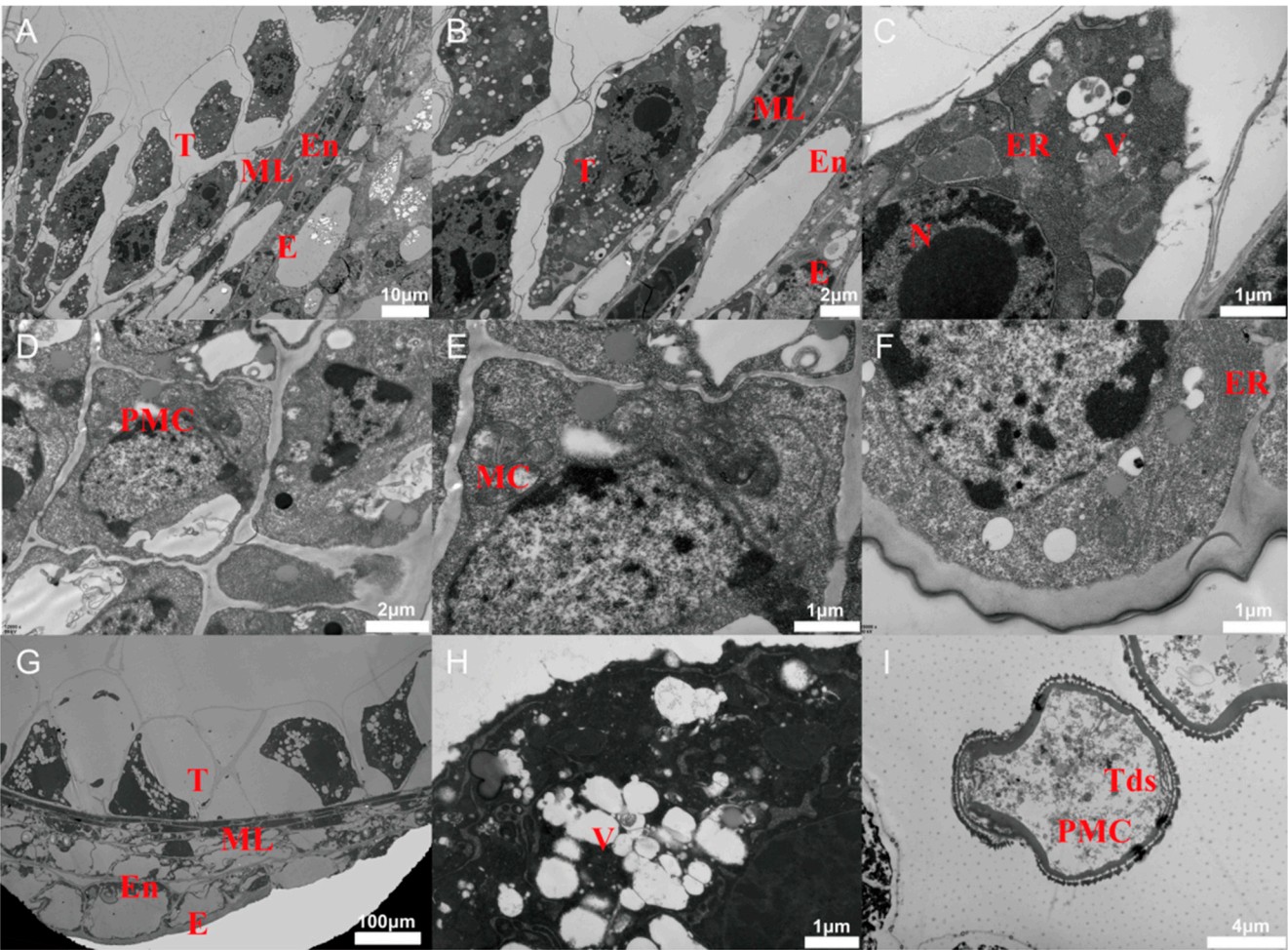

**Figure 2.** Anther ultrastructure of Ningqi No. 5 at Pm stage and Te stage. (**A**) Surface layer of pollen sac at Pm stage; (**B**) tapetal cell at Pm stage; (**C**) tapetal organelles at Pm stage; (**D**) pollen mother cell at Pm stage; (**E,F**) organelles in pollen mother cell at Pm stage; (**G**) anther surface layer at Te stage; (**H**) tapetal cells at Te stage; (**I**) pollen mother cell at Te stage. E, epidermis; EN, endothecium; ER, endoplasmic reticulum; MC, mitochondrion; ML, middle layer; N, nucleus; PMC, pollen mother cell; Po, pollen grain; SP, sporogenous cell; T, tapetum; Tds, tetrad; V, vacuole.

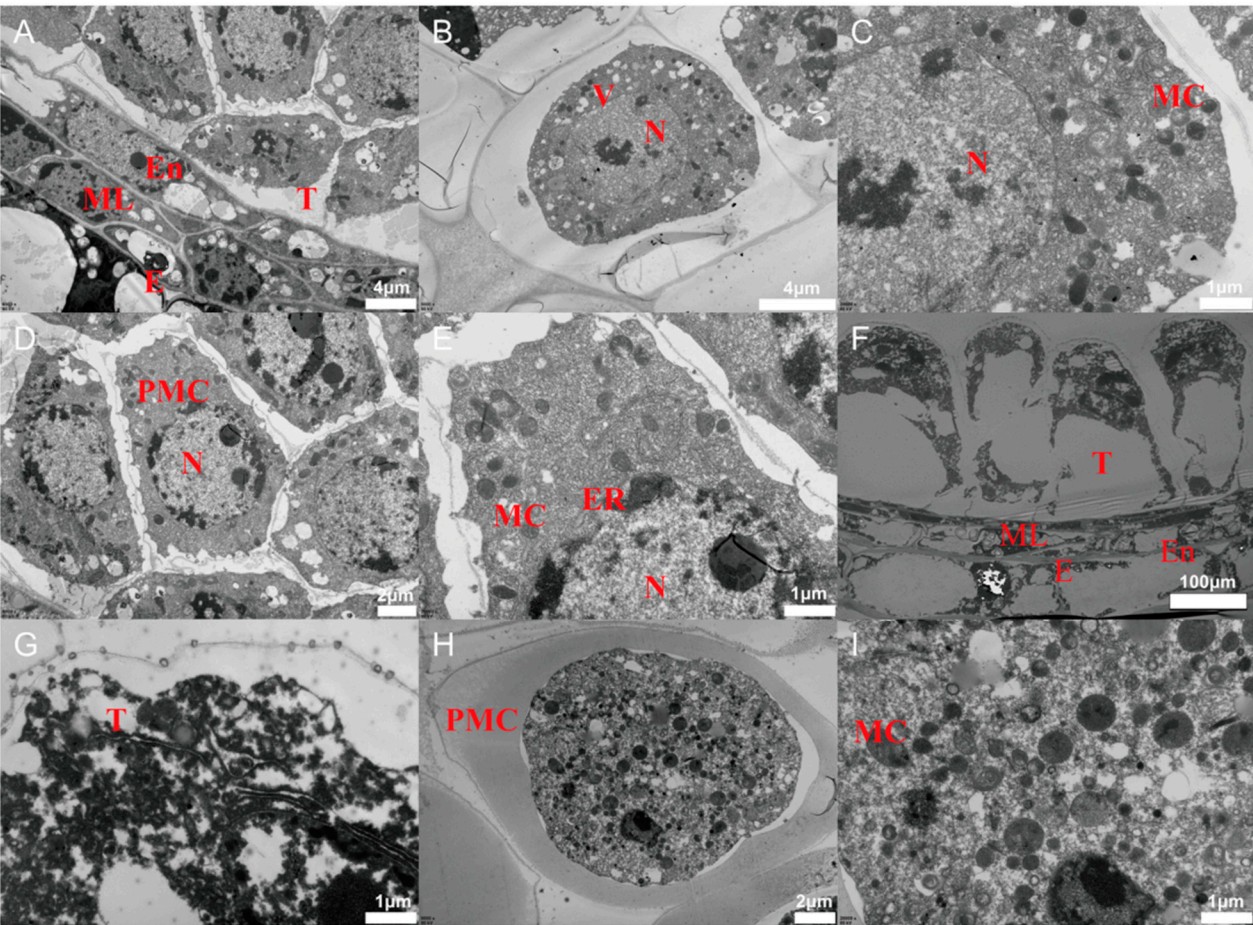

**Figure 3.** Anther ultrastructure of Ningqi No. 1 at Pm stage and Te stage. (**A**) Surface layer of pollen sac at Pm stage; (**B**) tapetal cell at Pm stage; (**C**) tapetal organelles at Pm stage; (**D**) pollen mother cell at Pm stage; (**E**) organelles in pollen mother cell at Pm stage; (**F**) anther surface layer at Te stage; (**G**) tapetal cells at Te stage; (**H,I**) pollen mother cell at Te stage. E, epidermis; EN, endothecium; ER, endoplasmic reticulum; MC, mitochondrion; ML, middle layer; N, nucleus; PMC, pollen mother cell; Po, pollen grain; SP, sporogenous cell; T, tapetum; Tds, tetrad; V, vacuole.

*3.2. PCD Signals*

Histocytological observation suggested that abnormal tapetal PCD may result in anther abortion in Ningqi No. 5. Considering that apoptosis is a sign of PCD, we measured the PCD state in the anther by detecting apoptotic signals. In the first two stages of anther development, no apoptosis signals were detected in either Ningqi No. 5 or Ningqi No. 1 by TUNEL (Figure 4A,B,F,G). At the Pm stage, PCD signals appeared in the tapetal cells of Ningqi No. 1 (Figure 4H), and relatively weaker signals were observed in the epidermis cells of Ningqi No. 5 (Figure 4C). At the Te stage, a weak apoptotic signal was found on the surface of Ningqi No. 1, and a strong signal was observed on the surface of the tetrad. This finding indicated that the callose surrounding the tetrad was degrading (Figure 4I), and the tapetum of Ningqi No. 5 was the strongest (Figure 4D). The apoptosis signal was weakened from the late Te stage to the Po stage but could still be detected in the tapetum with the same intensity as that in the microspores in the center of the pollen sac (Figure 4E). Ningqi No. 1 had only weak apoptotic signals from the Te stage to the Po stage (Figure 4J). This finding is consistent with the results that the apoptosis of tapetal cells in Ningqi No. 1 starts at the Pm stage when the tapetum is degraded to provide nutrients for microspore development. In Ningqi No. 5, the degradation of the tapetum starts at the Te stage, leading to insufficient nutrient supply in the early development of pollen mother cells, and thus,

pollen abortion. In summary, the delayed degradation of tapetal cells leads to microspore abortion in Ningqi No. 5.

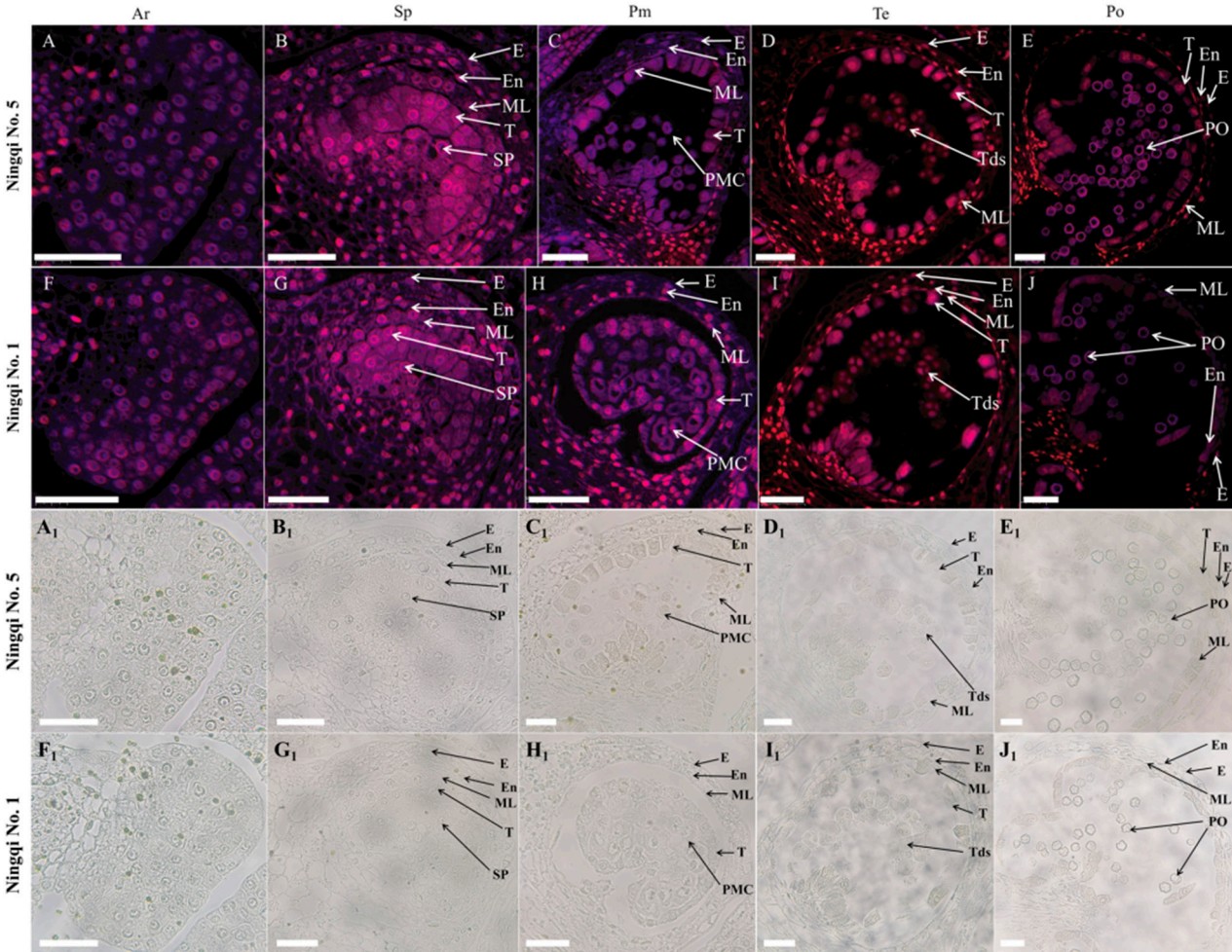

**Figure 4.** Apoptosis during pollen development of Ningqi No. 5 and Ningqi No. 1. (**A–E**) Pollen development of Ningqi No. 5; (**F–J**) pollen development of Ningqi No. 1; (**A₁–E₁**) and (**F₁–J₁**) are non-fluorescent photos of (**A₁–E₁**) and (**F₁–J₁**), respectively. (**A,F**) Pollen sac apoptosis at Ar stage; (**B,G**) pollen sac apoptosis at Sp stage; (**C,H**) pollen sac apoptosis at Pm stage; (**D,I**) pollen sac apoptosis at Te stage. (**E,J**) Pollen sac apoptosis at Po stage. E, epidermis; EN, endothecium; ML, middle layer; T, tapetum; Tds, tetrad; PMC, pollen mother cell; SP, sporogenous cell; Po, pollen grain. Red shows apoptosis signal. Scale bar = 50 μm.

*3.3. Contents of $H_2O_2$, $O_2^-$, and MDA in the Anthers of Ningqi No. 5 and Ningqi No. 1*

To determine whether oxygen free radicals are involved in microspore abortion, we analyzed the $O_2^-$ and $H_2O_2$ contents in anthers at five developmental stages. From the Ar stage, the $O_2^-$ content of Ningqi No. 5 and Ningqi No. 1 increased with microspore development and decreased after reaching the maximum level at the Pm stage. The $O_2^-$ content of Ningqi No. 5 was significantly higher than that of Ningqi No. 1 ($p < 0.01$) (Figure 5A). In the first three stages of anther development, the $H_2O_2$ content of Ningqi No. 5 and Ningqi No. 1 increased slowly, and no significant difference was observed between the two lines ($p = 0.9119$). The highest $H_2O_2$ content of Ningqi No. 1 was obtained at the Te stage, and that of Ningqi No. 5 was detected at the Po stage (Figure 5B). This finding indicated that ROS content is consistent with the period of anther abortion, and the accumulation of ROS is related to anther abortion.

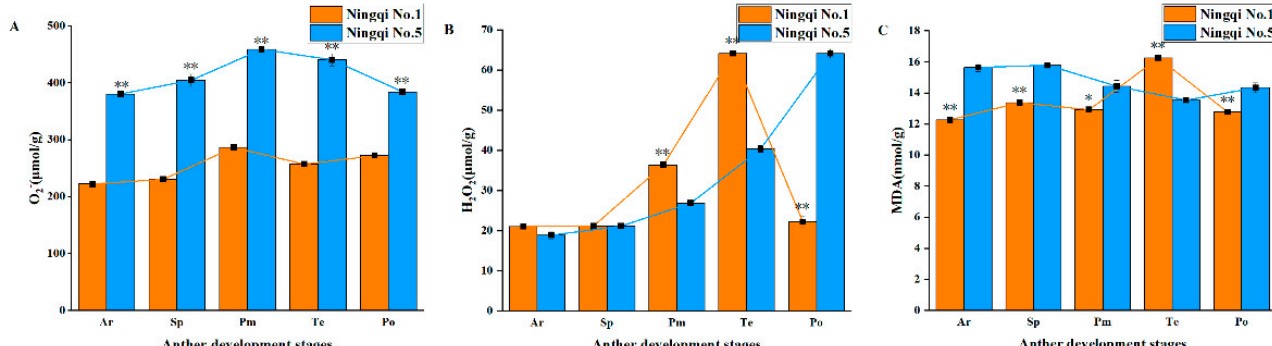

**Figure 5.** Contents of $O_2^-$, $H_2O_2$, and MDA during the pollen grain development of Ningqi No. 5 and Ningqi No. 1. (**A**) $O_2^-$ content; (**B**) $H_2O_2$ content; (**C**) MDA content. Ar, archesporial cell stage; Sp, sporogenous cell stage; Pm, pollen mother cell stage; Te, tetrad stage; Po, pollen grain stage. *, ** represents $p < 0.05$ and $p < 0.01$, respectively.

MDA is an indicator widely used for determining oxidative membrane damage and can directly reflect the degree of membrane damage. Ningqi No. 5 and Ningqi No. 1 both had an MDA burst stage. The MDA content peaked at the Te stage in Ningqi No. 1 when apoptosis was observed in the tapetum and at the Ar and Pm stages in Ningqi No. 5 (Figure 5C). The above results stated that in the progress of *L. barbarum* tapetal PCD, the oxidative membrane damage caused by ROS level is not synchronized with the initiation and development of tapetal PCD in Ningqi No. 1 and Ningqi No. 5.

### 3.4. Activities of Antioxidant Enzymes

In the first three stages of anther development, the GST activity was initially significantly higher in Ningqi No. 5 than in Ningqi No. 1, and then decreased in both lines (Figure 6A). The POD activity was significantly higher in Ningqi No. 1 than in Ningqi No. 5, and then changed to similar levels in both lines (Figure 6B). The same trend was observed for the APX activity in the two lines, but the APX activity was significantly higher in Ningqi No. 1 than in Ningqi No. 5 (Figure 6C). The SOD activity in both lines was enhanced at the Te stage, but the value in Ningqi No. 1 was higher than that in Ningqi No. 5 (Figure 6D). These results indicated that the antioxidant enzymes for scavenging ROS differ between the two lines. The main enzymes for scavenging ROS in Ningqi No. 1 are POD, SOD, and APX, and that in Ningqi No. 5 is GST. ROS accumulates during anther development, and Ningqi No. 1 can maintain the balanced state of free radicals by activating its antioxidant enzymes (POD, SOD, and APX). When ROS accumulates from the Ar stage to the Po stage in the sterile line, the ROS balance cannot be maintained, although the activities of antioxidant enzymes (GST and APX) increase slightly. This phenomenon results in abnormal tapetal PCD.

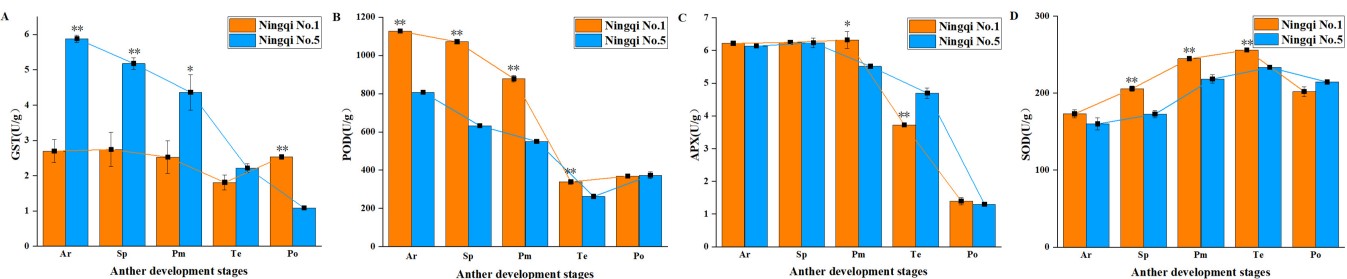

**Figure 6.** Antioxidant enzyme activities during the pollen development of Ningqi No. 5 and Ningqi No. 1. (**A**), GST activity; (**B**), POD activity; (**C**), APX activity; (**D**), SOD activity. Ar, archesporial cell stage; Sp, sporogenous cell stage; Pm, pollen mother cell stage; Te, tetrad stage; Po, pollen grain stage. *, ** represents $p < 0.05$ and $p < 0.01$, respectively.

### 3.5. Expression of Genes Involved in Regulatory Pathways of Tapetum Development

The qRT-PCR results showed that the *DYT1* expression level was significantly lower in Ningqi No. 5 than in Ningqi No. 1 from the Ar stage to the Po stage (Figure 7A). Meanwhile, the *MS1* expression level was higher in Ningqi No. 5 than in Ningqi No. 1 from the Ar stage to the Pm stage during pollen development. However, the *MS1* expression level decreased significantly in Ningqi No. 5 from the late Pm stage to the Te stage and eventually reached a level lower than that in Ningqi No. 1 (Figure 7B). Therefore, we speculate that the *DYT1-TDF1-AMS-MS188-MS1* pathway, which regulates tapetum development, affects the tapetum development of Ningqi No. 5 and is associated with pollen abortion.

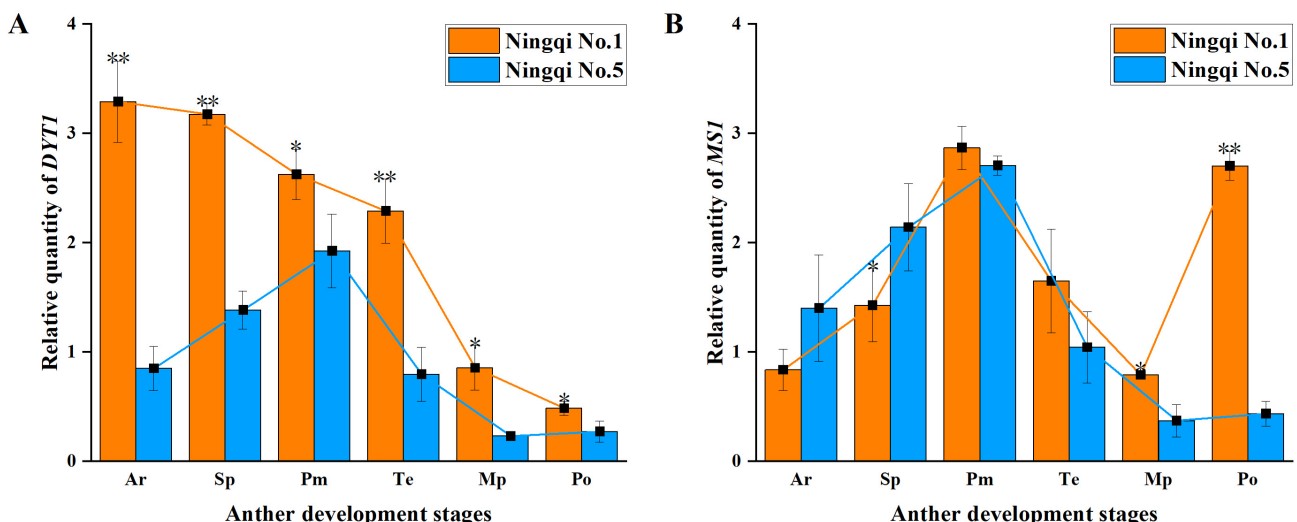

**Figure 7.** Relative quantity of *DYT1* gene (**A**) and *MS1* gene (**B**) during anther development in Ningqi No. 5 and Ningqi No. 1. (**A**) Relative quantity of *DYT1*; (**B**) relative quantity of *MS1*. Ar, archesporial cell stage; Sp, sporogenous cell stage; Pm, pollen mother cell stage; Te, tetrad stage; Mp, mononuclear pollen grain stage; Po, pollen grain stage. *, ** represents $p < 0.05$ and $p < 0.01$, respectively.

## 4. Discussion

### 4.1. Tapetum and Pollen Development

The tapetum plays a vital role in pollen development. Its normal development comprises timely PCD that provides nutrients for pollen development [30,31]. This study found that compared with that in the fertile line, the tapetum of the sterile line from the Pm stage to the Po stage was always close to the anther wall layer, did not sink into the pollen sac, and did not undergo PCD. Instead, the tapetum exhibited a persistent state. As a result, the pollen mother cells or microspores were spatially isolated from the tapetal cells during critical periods. Tapetum cells and male gametes originate from archesporial cells and completely enclose microspores during development. This close spatial relationship ensures the exchange of nutrients and structural or regulatory compounds during microspore development, including the release of exudates and the synthesis of various enzymatic fluids such as callose synthase and callosase [32]. The spatial isolation in the sterile line of *L. barbarum* prevents the enzyme solution secreted by the tapetal cells from reaching the pollen sac or promptly initiating PCD, thus, hindering the nutrient supply and leading to pollen abortion. The relationship between pollen abortion and tapetum development has been reported in many plants. A similar phenomenon was also observed in the sterile lines of Arabidopsis [33], rice [34], watermelon [35], and Brassica [36]. Therefore, the effect of abnormal tapetal PCD on the timely supply of nutrients is an important reason for the pollen abortion of the sterile line of *L. barbarum*.

In the ultra-electron microscope observation of Ningqi No. 5, the pollen mother cells began to exhibit abnormal mitochondrial morphology at the Pm stage—"Fzo," which was

first discovered in *Drosophila*. Fzo is the inability of sperm mitochondria to merge into large organelles, forming the nebenkern that resembles the cross-section of an onion [37]. Mitochondria undergo frequent fission and fusion in the cellular normal vital movement, and fission and fusion events are balanced. This phenomenon ensures the maternal inheritance of mitochondria and responses to new energy demands [38]. However, excess in fission and lack of fusion lead to mitochondrial network disruption, mtRNA loss, respiratory defects, and increased ROS [38]. In *Drosophila*, mitochondrial fusion is an important event during development, and it regulates spermatogenesis, specifically, by blocking mitochondrial fusion through mutations in the Fzo gene, resulting in male sterility [37]. Fzo has also been observed in plants, but its effect on reproductive development has not been explored. Numerous studies have demonstrated mitochondrial fission in cellular PCD [39]. In the current work, we found that Fzo specifically exists in the pollen mother cells of Ningqi No. 5 (Figure S1) and has possible effects on ROS metabolism, tapetal PCD, and pollen development.

### 4.2. ROS Level and PCD Progression

Tapetal PCD is associated with ROS level in cells [17,40–42]. ROS accumulates in the early stage of PCD but declines in the late stage of PCD [40,41]. High ROS levels can lead to abnormal PCD, which in turn causes abnormal tapetum function and pollen abortion [43,44]. In our study, low ROS levels appeared in the first two developmental stages of Ningqi No. 5 and Ningqi No. 1. In Ningqi No. 1, the ROS content increased at the Pm stage and then returned to a low level. However, the ROS content in Ningqi No. 5 increased in the late Te and Po period. The peak period of the ROS of the sterile line is one period behind that of the fertile lines. We speculate that when anther development enters the Te stage, ROS levels increase in the fertile lines. ROS act as signaling molecules to regulate the initiation of PCD, triggering signaling cascades and returning to a low level. Owing to various reasons, the dynamic ROS accumulation in Ningqi No. 5 does not initiate PCD until the late Te to the Po stage. As a result, the tapetum is preserved, which is also consistent with the discovery of Zhang [43] in rice.

### 4.3. Expression of Genes Regulating Tapetum Development and PCD and ROS Levels

Dynamic ROS balance is crucial for the expression of tapetum development-regulating genes and the initiation and progression of PCD [17,18]. Tapetal PCD-regulating gene *OE142* alters ROS metabolism in anthers, thereby affecting pollen fertility [45]. In tomatoes, the tapetum developmental gene *TDP1* regulates redox dynamics in anthers [46]. In this study, the expression levels of tapetum development genes *DYT1* and *MS1* significantly differed between the fertile and sterile lines. *DYT1* is expressed in anther development stage 4 in *Arabidopsis* and is mainly integrated in lipid metabolism, transport protein, and other physiological processes for pollen development [7]. *AtMYB103/MS188* is a member of the R2R3MYB gene family and is an upstream regulatory gene of the *MS1* gene. It not only plays a role in the tapetal development of anthers, but also plays a role in callose metabolism. Therefore, we conducted callose staining experiments on 10 flower buds at each of the five anther development stages of sterile and fertile lines [9]. The callose of the fertile and sterile lines in this study also significantly differed during anther development. In the early stage of the tetrad in the anther of Ningqi No. 1, the callose was thick around the tetrad but thinned out in the late tetrad. By contrast, the callose of Ningqi No. 5 was visible but significantly thinner than that of Ningqi No. 1 at the same stage. These trends are consistent with cytological observations and dynamic ROS levels (Figure S2, unpublished data). We speculate that the tapetum regulatory pathway is closely related to callose metabolism and ROS dynamic level. Although the regulatory cascade pathway remains unclear, further analysis will help us determine the molecular response signals to lay a foundation for a deep understanding of the relationship between ROS and tapetal PCD regulation.

## 5. Conclusions

The pollen abortion of Ningqi No. 5 started at the Pm stage and was induced by the preservation of tapetum. Based on all the results, we concluded that the dynamic level of ROS may be closely related to the development and regulation of tapetum in the sterile line of *L. barbarum*, ultimately leading to pollen abortion.

**Supplementary Materials:** The following supporting information can be downloaded at: https://www.mdpi.com/article/10.3390/horticulturae8111056/s1, Figure S1: Fzo in pollen mother cell at Pm stage; Figure S2. Callose development in Ningqi No.5 and Ningqi No.1 at different development stages: red is nuclear material, yellow is callose. (A, E). There was no significant change in callose in Pm stage in Ningqi No.1 and Ningqi No.5; B. There are clearly visible tetrads in the pollen sac cavity and thin callose outside the tetrads at prophase of Te stage in Ningqi No.5; C. Callose is not degraded at anaphase of Te stage in Ningqi No.5; D. At the Po stage, no microspore was found in Ningqi No.5; F. There are clearly visible tetrads in the pollen sac, and the callose wall around the tetrads is thick at prophase of Te stage in Ningqi No.1; G. Callose began degradation and becomes thin at anaphase of Te stage in Ningqi No.1; H. At the Po stage, round and deeply colored microspores can be seen in Ningqi No.1. Table S1: Main cytological characteristics of anther development in fertile and sterile lines; Table S2: List of primers used in this study.

**Author Contributions:** X.Z.: conceptualization, methodology, and writing—original draft preparation. Z.B.: software, data curation, and validation. H.M. and Z.W.: data curation and investigation. J.Z., Y.R., and W.X.: investigation and software. P.N., Y.W., and L.L.: methodology and software. W.Z., J.Y., and Y.Z.: methodology, visualization, and supervision. Z.S.: funding acquisition, supervision, and writing—reviewing and editing. All authors contributed to the article and approved the submitted version. All authors have read and agreed to the published version of the manuscript.

**Funding:** This study was supported by the Natural Science Foundation in Ningxia Province (2020AAC03245) and the Key Research and Development Plan of Ningxia Province (2021BEG03109).

**Institutional Review Board Statement:** Not applicable.

**Informed Consent Statement:** Not applicable.

**Conflicts of Interest:** The authors declare no conflict of interest.

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
