# Peer review of "Abnormal Programmed Cell Death of Tapetum Leads to the Pollen Abortion of Lycium barbarum Linnaeus"

_horticulturae, doi:10.3390/horticulturae8111056_

Round 1

Reviewer 1 Report

In this paper by Zhang and coworkers, the authors investigate the mechanisms determining the pollen sterility of a particular line of L. barbarum.

The analyses performed by the authors include: hystological observations, physiological charcterizations and transcriptomics.

Good things first: I found the general organization of the paper well structured and the topic interessting and with interessting future applications.

However, I also detected several issues that need to be fixed before the manuscript could be considered for the pubblication.

In general I think the statistics behind the presented data are poor and could be highly improved. In more details:

1- abount the analyses of microscopic images: how many images did the author analyzed? how the authors analyzed the images? please clarify the parameter analyzed and how they have been measured.

2- the method section is generally poor and lacks references. for example about the MDA assay. how does it work? which protocl did you used?

Figure 4: A non fluorescent image need to be added to clarify what we are looking at, and what we are comparing.

Figure 5 and 6: these images are very hard to understand. maybe a more clear 2D representation would help readers in the interpretation

Figure 7: the error bars in the graphs are huge and the statistics lacks. I wuold suggest to repeat the experiment with an higher number of samples  and perform ANOVA/T-test on the results; otherwise all the output form this analysis can't be informative. 

Figure S1 and S2: what we are looking at? how many images have been analyzed? how did the author performed the measurments and the observations? 

General comment:

I thinks the potential of this paper is good, however the general quality of the results and presentation need to be enhanced. 

As the authors themselves states all the results descibed in the paper are highly specultative, and this is mostly caused by the bad statistics and the low number of samples analysed. 

Reviewer 2 Report

Dear Authors,

Reviewer comments horticulturae-1967456

The manuscript entitled „Abnormal programmed cell death of tapetum leads to the pollen abortion of Lycium barbarum L.“ represents a useful study aimed at a microscopic investigation of pollen development with a focus on tapetum and the role of programmed cell death (PCD), ROS levels and ROS scavenging enzymes activities in microspore development. PCD in tapetum layer provides nutrients for pollen development. In the sterile line, PCD of the tapetum layer was significantly delayed which was associated with altered expression of DYT1 and MST genes involved in regulation of tapetum development. The results of the study indicate that abnormal tapetal PCD can underlie male / pollen sterility in Lycium barbarum.

I can recommend the manuscript for publication in Horticulturae.

However, I have a few comments on the present manuscript which are given below:

Figure 4: In Figure 4, the details on pollen development are poorly visible on the microphotographs. I think that the quality of the microphotographs has to be improved.

In Figure 5 and Figure 6 showing the 3D graphs, the axis headings are small and cannot be read in the present format. I have not found what is shown by the second horizontal axis – y-axis – in all graphs in both Figure 5 and Figure 6. No information on the second horizontal axis is given in the figure legends. Only the information on the vertical axis (z-axis) showing eitehr ROS contents (Fig. 5) or ROS scavenging enzyme activities (Fig. 6) and the first horizontal axis (y-axis) showing the stages of pollen development are given in the figure legends. I thus think that the information on the second horizontal axis including the characterististics shown by the axis has to be added to both figure legends. Moreover, in Figure 6 legend, i tis written that antioxidant enzymes activities are shown which corresponds to the data given in Materials and methods while further in the legend, we can red about GST content, POD content, APX content, and SOD content. However, the authors have to keep in mind that there is a big difference between enzyme content, i.e., relative abundance of the protein, and enzyme activity, i.e., the amount of the product made by the enzyme per unit of the time, so these two terms are different and cannot be used synonymously. I think that based on the information in Materials and methods, the authors just determiend enzyme activities, not enzyme (protein) content and thus the figure legend has to be corrected.

Formal comments on the text:

Materials and methods, part 2.7. Correct the typing error in the name „Trizol reagent“ (not „Triol reagent“).

Discussion, page 9, the last line in the first paragraph in Discussion section: Write plant scientific name „L. barbarum“ in italics.

Final recommendation: Accept after a minor revision.

Round 2

Reviewer 1 Report

I found this second version of the ms highly improved. I really appreciate the efforts of the authors to respond to all my comments.

The statistics are now clear and well conducted and the number of samples adequate.

I just have a couple of minor comments:

1- In figure 7. in the Y axe and in the caption the right definition is "Relative Gene Expression" or "Amount of transcript" not "GENE QUANTITY"

2- please correct all the species name in the ms using italic font.

3- thanks for the "response 6" in the previous revision round, now it is much clear to me, however I would suggest to add some words in the discussion or in the conclusions about that.

4- Again I would say that the results are a little bit speculative so I would suggest to state it more clearly in the conclusions. 
